# Transcriptome Sequencing of *Broussonetia papyrifera* Leaves Reveals Key Genes Involved in Flavonoids Biosynthesis

**DOI:** 10.3390/plants12030563

**Published:** 2023-01-26

**Authors:** Peng Guo, Ziqi Huang, Xinke Li, Wei Zhao, Yihan Wang

**Affiliations:** 1College of Life Science, Henan Agricultural University, Zhengzhou 450046, China; 2Henan Engineering Research Center for Osmanthus Germplasm Innovation and Resource Utilization, Zhengzhou 450046, China

**Keywords:** *Broussonetia papyrifera*, flavonoids, RNA-seq

## Abstract

*Broussonetia papyrifera* is rich in flavonoids, which have significant antioxidant, antibacterial, and anti-inflammatory activities and certain pharmacological activities. Nevertheless, scarce transcriptome resources of *B. papyrifera* have impeded further study regarding the process of its production and accumulation. In this study, RNA-seq was utilized to evaluate the gene expression of *B. papyrifera* leaves at three distinct developmental phases (T1: young leaves, T3: immature leaves, T4: matured leaves). We obtained 2447 upregulated and 2960 downregulated DEGs, 4657 upregulated and 4804 downregulated DEGs, and 805 upregulated and 484 downregulated DEGs from T1 vs. T3, T1 vs. T4, and T3 vs. T4, respectively. Further research found that the following variables contributed to the formation of flavonoids in the leaves of *B. papyrifera*: Several important enzyme genes involved in flavonoid production pathways have been discovered. The results demonstrated that the dynamic changing trend of flavonoid contents is related to the expression pattern of the vast majority of essential genes in the biosynthetic pathway. Genes associated in energy and glucose metabolism, polysaccharide, cell wall and cytoskeleton metabolism, signal transduction, and protein and amino acid metabolism may affect the growth and development of *B. papyrifera* leaves, and eventually their flavonoid content. This study’s results offer a strong platform for future research into the metabolic pathways of *B. papyrifera*.

## 1. Introduction

*Broussonetia papyrifera* (L.) Vent. is a perennial tree of *Broussonetia* family in Moraceae. *B. papyrifera* is unisexual and dioecious, combining the fundamental plant development characteristics of fast growth, a broad appropriate range, and a high fiber content. The whole plant of *B. papyrifera* was used as medicine and included in the compendium of Materia Medica, from which 56 flavonoids such as broussoflavonol, alkaloids, and coumarins were identified [1].

Flavonoids are a kind of natural compound with a 2-phenylchromone structure, portraying significant plant functions, including growth, development, flowering, fruit-bearing, antibacterial therapy, and disease prevention. They can also be used as additives in functional foods as natural antioxidants, natural pigments, and natural sweeteners. *B. papyrifera* mainly includes broussoflavone, broussoflavonol, kazinol, and other flavonoids. Total flavonoids were the main effective components for inhibiting atrial contractility [2], have antioxidant, antibacterial, and anti-inflammatory activities for treating breast cancer, and have a protective effect against the oxidative damage to human-immortalized epidermal cells poisoned by lead and arsenic [3].

The biosynthesis of flavonoids mainly involves the phenylpropane metabolic pathway and the flavonoid metabolic pathway. It requires the joint action and participation of phenylalanine ammonia-lyase (PAL), cinnamate 4-hydroxylase (C4H), 4-coumarate CoA ligase (4CL), chalcone synthase (CHS), flavonol synthase (FLS), and other enzymes [4,5]. As part of the phenylpropane metabolic route, PAL converts phenylalanine to cinnamic acid. Cinnamic acid is next transformed to p-coumaric acid by C4H, which catalyzes the introduction of hydroxyl on the benzene ring. The carboxyl group of p-coumaric acid is subsequently activated through a thioester bond catalyzed by 4CL to generate p-coumaric coenzyme A (COA) [6]. After that, 4-coumarinyl coenzyme A condenses with three malonyl coenzymes to form chalcone through CHS [7]. CHS belongs to the type III polyketide synthase superfamily and exists in most plants [8]. The mechanism of CHS action shows that, as the starting molecule, 4-coumarinyl coenzyme A binds to cysteine residues in the CHS active site and then forms tetraketone intermediates through a series of decarboxylation condensation reactions, including the addition of three malonyl coenzyme A supplement molecules. Then, chalcone (4, 2′, 4′, 6′- tetrahydroxychalcone) is produced by intramolecular cyclization of tetraketone intermediates [8]. Chalcone isomerase (CHI) converted it to a flavanone (like naringenin or eriodictyol) [9]. The last step in the formation of flavonoids is the introduction of a double bond between the C-2 and C-3 sites by flavone synthase (FNS) [10]. As part of the soluble Fe^2+^/2-oxoglutarate dependent dioxygenase (2-ODD) family, FNSI catalyzes flavanone conversion in order to become flavonoids [10]. The first FNSI enzyme was identified in the leaves of parsley. The assessment of its activity indicated that FNSI transformed 14C-radiolabeled flavanone into equivalent flavonoids with no detectable intermediates [11]. After these preliminary studies, FNSI enzymes were identified in many species of Apiaceae [12,13]. A recent study has shown that the rice FNSI enzyme OsFNSI-1 converts flavanone (2S)-naringin to apigenin in vitro (Lee et al., 2008), suggesting that 2-ODD with FNS activity is more prevalent than previously believed. The majority of FNSII enzymes are oxygen- and NADPH-dependent, membrane-bound cytochrome P450 (CYP) monooxygenases found in higher plants [14]. All known FNSII enzymes belong to either the dicotyledon CYP93B subfamily or the monocotyledon CYP93G subclass. Most FNSII enzymes convert flavanones directly into flavonoids by creating a double bond between the C-2 and C-3 residues of flavanone [14].

Multiple studies have indicated that the expression of PAL, C4H, 4CL, CHS, FLS, and other genes is easily affected by external factors [15,16,17,18,19]. In addition, the synthesis of flavonoids is also regulated by transcription factors. BHLH transcription factors belong to the helix-turn-helix (HLH) type of transcription factors, whose domains are highly conserved among species and are composed of basic regions and helix-turn-helix, which can regulate the synthesis of flavonoids [20]. The MYB transcription factor is a huge transcription factor in plants. Its N-terminal has a relatively conserved helix-turn-helix domain composed of 50–53 amino acids, and its C-terminal is not conserved. Significant differences exist between different or the same species [21]. The MYB transcription factor can regulate the synthesis of flavonoids in apples, snapdragons, tomatoes, lilies, soybeans, narcissus, and other plants. The WD40 transcription factor family is a protein with 4–16 tandem repeats of the WD domain composed of about 40 amino acids [22]. WD-40 protein itself has no catalytic function. It usually interacts with the MYB and bHLH families through its WD domain to stabilize the complex and form the MYB-bHLH-WD40 protein complex, namely the MBW protein complex, which jointly acts on the anthocyanin synthesis pathway [23]. In bayberry, MrWD40-1 can form an MBW complex with MrbHLH1 and MrMYB1, thereby regulating the synthesis and accumulation of anthocyanin [24], and there are similar conclusions in pomegranate [25]. With the development of high-throughput sequencing technology in recent years, transcriptome sequencing (RNA-Seq) has become the dominant approach for analyzing gene expression regulation and has been extensively used to the identification of novel genes, the determination of metabolic pathways, the mining of molecular markers, gene family identification, and evolutionary analysis. Various genes associated with the biosynthesis and regulation of flavonoids were successfully screened by RNA-Seq technology in economic plants.

Therefore, this study intends to use the leaves of *B. papyrifera*, rich in flavonoids, as materials to determine the content of flavonoids and the expression of regulatory genes in the leaves of *B. papyrifera* at different growth stages. This will further reveal the biosynthesis pathway and regulation mechanism of flavonoids in *B. papyrifera*, lay the foundation for cloning key genes and functional analysis, and provide a theoretical basis for the formation of *B. papyrifera* quality.

## 2. Results

### 2.1. Changes in Total Flavonoid Content during B. papyrifera Leaves Development

The content of flavonoids gradually increased from the T1 (young leaves) period to the T4 (matured leaves) period with notable variations at different times (Figure 1).

### 2.2. RNA-Seq, De Novo Assembly of the Transcripts and Annotation

T1 (T1-1, T1-2, and T1-3), T3 (T3-1, T3-2, and T3-3), and T4 (T4-1, T4-2, and T4-3) *B. papyriferas* libraries, each with three biological replicates, were constructed and sequenced. The number of raw reads produced by each library varied between 21,308,885 and 23,421,640. The percentages of clean reads and Q20 (rates of sequencing errors less than 1%) were more than 95% and 97%, respectively (Table 1). The Trinity program produced 41,537 unigenes with lengths ranging from 301 to 17,100 bp (average length: 1515 bp) and a N50 length of 2568 bp. Further, 11,342 (27.31%) of these unigenes were 300–500 bp in length, 10,629 (25.59%) were 501–1000 bp in length, 7915 (27.52%) were 1–2 kb in length, and 11,651 (28.05%) were >2 kb in length (Table 2 and Table 3).

### 2.3. Functional Annotation and Classification

All 41,537 constructed unigenes were successfully annotated to the seven databases using the BLAST method. Annotation was performed on 30,177 unigenes, or 72.65% of the total. All seven databases annotated 3358 (8.08%) unigenes effectively. Of the 41,537 unigenes, the highest annotation success rate in the database was Nr 62.12%, followed by Swiss-Prot 53.81%, Pfam 51.25%, GO 51.24%, Nt 47.15%, KO 24.94%, and KOG 20.41% (Table 4).

The classification of 21,287 unigenes into biological process (BP), cellular component (CC), and molecular function (MF) was accomplished. In terms of BP and CC, these unigenes were assigned 25 and five GO keywords, with ‘cellular process’ (12,694) and ‘cellular anatomical entity’ (9419) being the most prevalent subcategories. These matching unigenes were assigned 12 GO keywords in the MF category, with ‘binding’ (11,698) and ‘catalytic activity’ (9605) being the most prevalent (Figure 2). The KOG analysis identified 8480 unigenes categorized into 25 categories: O (Posttranslational modification, protein turnover, chaperones), R (General function prediction alone), and J (Translation, ribosomal structure, and biogenesis) (Figure 3). 8844 unigenes were attributed to 22 metabolic pathways in the KEGG database. These 22 KEGG pathways were categorized into five groups: cellular processes (725 unigenes), environmental information processing (1273 unigenes), genetic information processing (2397 unigenes), metabolism (3915 unigenes), and organismal systems (534 unigenes). The metabolic pathway with the greatest number of unigenes was ‘signal transduction’ (1202), followed by ‘Translation’ (995) (Figure 4).

BLASTp sequence comparisons identified the unigenes to known TF gene families (PlantTFDB: http://planttfdb.gao-lab.org/, accessed on 25 May 2021). According to our results, a total of 1728 transcription factors (TFs) found in *B. papyrifera* may be categorized into 87 TF families (Appendix A). Among them were the transcription factor (TF) families bZIP, v-Myb avian myelo-blastosis viral oncogene homologue (MYB), and basic helix-loop-helix (bHLH). The most prevalent of these TFs was C2H2 (147), followed by bHLH (83) and AP2/ERF-ERF (74). In addition, DBP, STAT, C2C2-CO-like, S1Fa-like, AP2/ERF-RAV, and LFY were least represented in the *B. papyrifera* transcriptome.

### 2.4. DEG Identification

Using an absolute value of the log2 ratio ≥ 1 and an FDR ≤ 0.05 threshold, we got 2447 upregulated and 2960 downregulated, 4657 upregulated and 4804 downregulated DEGs, and 805 upregulated and 484 downregulated DEGs from T1 vs. T3, T1 vs. T4, and T3 vs. T4, respectively (Figure 5a).

As seen in Figure, we identified 9989 DEGs between T1 and T3, T1 and T4, and T3 and T4, with only 610 DEGs shared by the two groups (Figure 5b). Changes in TFs and genes involved in cellular transport, polysaccharide, cell wall, and cytoskeleton metabolism, carbohydrate and energy metabolism, protein and amino acid metabolism, lipid metabolism, signal transduction, and hormone metabolism varied between T1, T3, and T4 in response to flavonoid biosynthesis. The flavonoid production and gene expression patterns of leaves from various seasons varied significantly. Hence, 59.00% of DEGs had an absolute value of log2 the ratio < 2 in T1 vs. T3, whereas only 45.15% of DEGs in T1 vs. T4 and 67.80% of DEGs in T3 vs. T4, indicating that as leaves grew, gene expression varied more in T1 vs. T4 than in T1 vs. T3 (Table 5).

GO enrichment analysis was carried out on DEGs in the three groups (T1 vs. T3, T3 vs. T4, T1 vs. T4), In the T1 vs. T3 group, DEGs were mainly concentrated in “DNA binding,” “oxidoreductase activity,” “non-membrane-bounded organelle,” and so on. In the T3 vs. T4 group, “catalytic activity,” “transferase activity,” and “phosphotransferase activity,” with alcohol acting as an acceptor, were mainly enriched. In the T1 vs. T4 group, it was primarily enriched in, among other things, “metabolic process,” “biosynthetic process,” and “cellular biosynthetic process.” This suggests that DEGs with these functions may be essential for the formation and growth of *B. papyrifera* (Figure 6).

The study of KEGG enrichment was then conducted for each DEG category. Figure 7 illustrates the top 20 significantly enriched pathways. KEGG enrichment analysis found that “Photosynthesis” was the most substantially enriched metabolic pathway in group T1 compared to group T3, followed by “DNA replication” and “Cell cycle.” In the T3 vs. T4 group, there were 215 metabolic pathways, and the KEGG enrichment analysis indicated that “Starch and sucrose metabolism” was the most substantially enriched pathway, followed by “Biosynthesis of amino acids.” The KEGG enrichment analysis found that “Ribosome” was the most substantially enriched metabolic pathway in the T1 vs. T4 group, followed by “Plant hormone signal transduction” and “Photosynthesis.” These results indicate that the development of *B. papyrifera* leaves includes a sequence of secondary metabolite modifications and plant hormone signal transduction.

### 2.5. Validation of qRT-PCR

The results of all DEGs obtained by the qRT-PCR analysis were highly related to the data produced by RNA-Seq (Figure 8 and Appendix A). Thus, the RNA-Seq data were reliable.

## 3. Discussion

### 3.1. Putative Genes Related to Flavonoid Biosynthesis and Transport

The phenylpropanoid metabolic pathway results in the production of flavonoids. The transcriptome of *Broussonetia papyrifera* included 141 putative unigenes involved with the phenylpropanoid biosynthesis pathway (ko00940). In addition, the transcriptome of *B. papyrifera* revealed 41 candidate unigenes linked with the flavonoid biosynthesis route (ko00941) and five candidate unigenes connected with the flavone and flavonol biosynthesis pathways (ko00944).

Flavonoids are natural compounds with a 2-phenylchromone structure found in plants. At least 56 flavonoids, such as broussone, alkaloids, and coumarins, were identified in *B. papyrifera* [1]. The biosynthesis of flavonoids mainly involves phenylpropanoid and flavonoid pathways. In the transcriptome of *B. papyrifera*, putative flavonoid structural genes included the coding of genes for phenylalanine ammonia lyase (PAL, 2), 4-coumarate CoA ligase (4CL, 4), chalcone synthase (CHS, 4), chalcone isomerase (CHI, 2), flavonoid 3-hydroxylase (F3H, 1), flavonoid 3-hydroxylase (F3′H, 2), dihydro- flavonol 4-reductase (DFR, 1), anthocyanidin synthase (ANS, 1), Leucoanthocyanidin reductase (LAR, 1), and anthocyanidin reductase (ANR, 1). This study determined 32 cytochrome P450 (CYP) and 20 O-methyltransferase (OMT) unigenes. Hence, these genes may be involved in the alteration of flavonoids in *B. papyrifera*. It has been shown that flavonoid modification involves CYP and OMT [26]. Hydroxylation and methylation are crucial modifying events that have the ability to dramatically expand flavonoids’ chemical diversity [27].

The results demonstrated that the expression levels of essential enzyme genes in flavonoid biosynthetic pathway 2 PAL (Cluster-10108.1150, Cluster-10108.1151), 4 4CL (Cluster-10108.7191, Cluster-10108.4707, Cluster-10108.2772, Cluster-10108. 10116), 2 CHI (Cluster-10108.22335, Cluster-10108.19098), 1 LAR (Cluster-10108.5296), and 1 DFR (Cluster-10108.7225) were upregulated in T1 vs. T4. Meanwhile, the expression levels of 1 4CL (Cluster-10108.14282), 1 CYP73A (Cluster-10108.10698), 1 ANS Cluster-10108.12273, 1 F3H (Cluster-10108.9733), 4 CHS (Cluster-10108.9613, Cluster-10108.13424, Cluster-10108.20907), Cluster-10108.4110), and 2 F3′H (Cluster-10108.2078, Cluster-10108.14936) were downregulated in T1 vs. T4 (Appendix A). Among these key genes, seven genes were downregulated, two genes were upregulated in T1 vs. T3, one gene was up-regulated and one was down-regulated between T3 and T4. The bulk of the aforementioned gene expression patterns were comparable to the buildup of total flavonoid content in *B. papyrifera* leaves as determined by UV spectrophotometry, showing that they were closely associated with flavonoid production (Figure 9). Moreover, we ran a cluster analysis on the expression levels of the same gene in different samples and the expression patterns of several genes in the same sample (Figure 10). T3 and T4 were gathered into one class, whereas T1 was clustered into another. T3 and T4 were more closely connected to one another than T1. All genes with down-regulated expression at the T4 stage grouped together, while all genes with up-regulated expression at the T4 stage clustered together. In the second group, the majority of DEGs belonged to the route upstream of phenylpropanoid production.

Based on previous studies and according to the different specific transport mechanisms of flavonoids, some scholars have proposed three mechanisms of flavonoid transport in plant cells: vesicle trafficking, membrane transporters-mediated flavonoid transport, and glutathione S-transferase [28]. There are mainly two membrane proteins involved in flavonoid transport: the ABC (ATP-binding cassette) protein family and the MATE (multidrug and toxic compound extrusion) protein family. Many flavonoid transport-related genes were isolated from the *B. papyrifera* transcriptome. The transcriptome data revealed 398 unigenes that code for ABC transporters, 64 unigenes that code for glutathione S-transferase (GST), eight unigenes that code for H+-ATPases, and five unigenes that code for vacuolar sorting receptors (VSRs). In addition, it was discovered that three ABC transporter unigenes, one glutathione S-transferase unigene, and two H+-ATPase unigenes were highly up-regulated. These upregulated unigenes in *B. papyrifera* may be involved in the transfer of flavonoids from the cytosol to the vacuole. In addition, two glutathione S-transferase unigenes were discovered to be highly down-regulated, which may have important implications.

### 3.2. Genes Related to Signal Transduction

Plant hormones play a significant role in the accumulation of flavonoids, which has been clarified in previous studies. The genes involved in hormone production and signal transduction were compared in this research. Two DEGs implicated in the BR synthesis pathway were identified, one of which, BR6OX1, was downregulated either in T1 vs. T3 or in T1 vs. T4. In contrast, the BAS1 gene, which decreases the amount of active BRs, was indistinguishably downregulated in T1 vs. T4. In addition, just one DEG implicated in the GA synthesis pathway, KAO, was discovered to be up-regulated in the leaves of *Broussonetia papyrifera*. In *B. papyrifera*, practically all of the DEGs implicated in JA biosynthesis, including five AOCs and one OPR, were upregulated. These expressions contributed to the enhancement of the synthesis pathways for BR, GA, and JA.

MeJA is an essential phytohormone for plant development and several physiological and biochemical activities [29]. MeJA increases the antioxidant capacity and flavonoid concentration of blackberries, strawberries, and olive fruits [30,31,32]. In this study, five of seven DEGs related to JA synthesis were upregulated in *B. papyrifera*. Jasmonate Zim domain (*JAZ*) protein, an inhibitor of the jasmonate signal transduction pathway, can participate in flavone metabolism pathway as a repressor protein. It can directly interact with MYB transcription factors, and then MYB acts on corresponding target genes to affect the flavone metabolism pathway (Sun et al., 2014). In the previous studies, the content of total flavonoids in the leaves of tartary buckwheat sprouts treated with MeJA increased. The correlation analysis showed that the content of total flavonoids was significantly positively correlated with *ftJAZ* [33]. We isolated three downregulated and one upregulated JAZ from *B. papyrifera* leaves. Therefore, this may be one of the reasons for the change of flavonoids in *B. papyrifera*.

Additionally, a number of DEGs engaged in seven hormone signal transduction pathways were discovered, including nine in ABA, 30 in AUX, five in BR, 10 in CTK, one in GA, 13 in JA, and four in SA. Seven ABA signaling DEGs were downregulated, representing the vast majority of AUX signaling pathway genes whose expression was downregulated in *B. papyrifera*. Further, three out of five DEGs involved in BR signaling were upregulated in *B. papyrifera* (Appendix A); nine of the ten DEGs involved in the JA signal pathway, and seven out of ten DEGs involved in CTK signaling were upregulated. Therefore, the genes related to signal transduction and hormone metabolism changes might affect the content of flavonoids in *B. papyrifera* leaves and their growth and development.

### 3.3. Transcription Factors

As demonstrated in Appendix A, 325 downregulated and 344 upregulated TFs were identified in the leaves of *Broussonetia papyrifera*. The vast majority belonged to the myeloblastosis (MYB), Apetala2 (AP2)/ethylene-responsive transcription factor (ERF), zinc finger (C-x8-C-x5-C-x3-H (CCCH) and Cys2/His2 (C2H2)), WRKY, helix-loop-helix (bHLH), and NAC families. Thus, the obtained transcriptome data may be utilized to identify regulatory networks involved in flavonoid metabolites and other metabolites in the leaves of *B. papyrifera*.

MYB transcription factor has a wide range of functions that involve almost all aspects of plant development and metabolism. In recent years, there have been more and more studies on the accumulation of anthocyanins by the MYB protein in the flavonoid synthesis pathway of plants. For example, MYB transcription factors in apples, strawberries, grapes, and other plants have been cloned one after another [21,34,35]. Here, we identified 37 upregulated and 25 downregulated MYBs in *B. papyrifera* leaves. BHLH (basic helix loop helix) is the second largest transcription factor family among plants. It plays a vital role in regulating many physiological pathways in plants. One of the most significant functions of bHLH transcription factors is to regulate flavonoid production [36]. In the leaves of *B. papyrifera*, we found 23 upregulated and 19 downregulated bHLHs. A MYB-interacting region (MIR) structure in maize is crucial for color production and is triggered by the interaction between bHLH and R2R3-MYB proteins [36]. Thus, bHLH TFs may interact with MYB proteins in our research, regulating flavonoid production. WD40 is also a transcription factor family related to regulating flavonoid synthesis genes in plants. We identified 27 upregulated and three downregulated WD40s in *B. papyrifera* leaves. In addition, the transcription factors (TFs) bHLH, MYB, and WD are present in the majority of plant species. They can regulate flavonoid production and transport at the transcriptional level as MBW ternary complexes [37]. Our results provide clues for revealing the gene regulatory network of flavonoid synthesis in *B. papyrifera*. To identify the particular roles of these hypothetical TFs, more study is required.

### 3.4. Other Genes That May Affect the Synthesis of Flavonoids in Broussonetia papyrifera

Energy shortages affect the growth and development of leaves. A close relationship exists between the content of flavonoids and the growth and development of leaves. We isolated six downregulated genes and 10 upregulated as a part of the triad of essential enzymes (hexokinase, phosphofructokinase, and pyruvate kinase) within the glycolysis pathway from the *B. papyrifera* transcriptome. Thus, the higher glycolysis might contribute to the growth and development of leaves, meeting the requirement of leaves for energy. The products of the glycolysis pathway were subsequently catalyzed by the pyruvate dehydrogenase complex and incorporated into the TCA cycle. During the growth of *B. papyrifera* leaves, two pyruvate dehydrogenase genes were up-regulated, whereas one citrate synthase gene rose and one died. Two succinate dehydrogenase genes were up-regulated in *B. papyrifera*, whereas three succinate dehydrogenase genes were down-regulated. In addition, five of the seven malate dehydrogenase genes were down-regulated in the leaves of *B. papyrifera*. In conclusion, the research identified a greater number of up-regulated (439) than down-regulated (361) genes associated with energy and carbohydrate metabolism from *B. papyrifera* leaves, which might indicate that the energy required increased gradually with the growth and development of leaves (Appendix A).

Protein and amino acids play essential roles in plant development because plant cells mainly perform their complex physiological functions through proteins. *Broussonetia papyrifera* leaves are rich in protein, especially the content of free amino acids in young leaves, in which glutamate is high [38]. We isolated six downregulated and three upregulated genes related to glutamate amino acid synthesis from *B. papyrifera* leaves, which might be related to the growth and development of *B. papyrifera* leaves. Protein modification is the crucial mechanism of plant proteome diversity, and phosphorylation is the most common PTM in plants [39]. A DEG study of amino acid metabolism pathways indicated that protein phosphorylation in *B. papyrifera* leaves was nearly totally downregulated. In plants, amino acids perform crucial functions. We identified 293 upregulated and 247 downregulated protein and amino acid metabolism-related genes from *B. papyrifera* leaves (Appendix A), suggesting that amino acid metabolism may play a role in the growth and development of *B. papyrifera* leaves.

We identified 115 downregulated and 184 upregulated polysaccharide and cell wall metabolism genes between T1 and T4, suggesting that polysaccharide and cell wall metabolism increased progressively with leaf growth and maturation. In previous studies, the number of callose layers wrapped outside the cell wall increased in growing cells, which reduces the elasticity of the cell wall, inhibits the growth of cells, and affects the growth and development of plants. The downregulation of callose synthase in *B. papyrifera* leaves supported the above inference. The development of plant cells requires alterations to the cellulose-xyloglucan network (primary cell walls) [40]. The alteration of primary cell walls is catalyzed by a number of enzymes, including the xyloglucan endotransglucosylase/hydrolase (XTH) family protein. XTHs catalyze the endotransglycosylation of xyloglucan by means of xyloglucan endotransglucosylase (XET) activity and/or the hydrolysis of xyloglucan by means of xyloglucan endohydrolase (XEH) activity, both of which contribute to cell wall degradation [41]. Here, we identified four downregulated and twelve upregulated XTH(XET) genes from *B. papyrifera* leaves, suggesting that the upregulation of XTH(XET) played a role in the leaf growth of *B. papyrifera*. Expansins, which function as wall-loosening proteins, promote plant development [42]. As shown in Appendix A, we isolated four downregulated and eight upregulated expansins, six upregulated and one downregulated extensins from *B. papyrifera* leaves. It has also been suggested that ascorbate oxidase (AO) promotes cell elongation by generating monodehydroascorbate and dehydroascorbate [43]. Five upregulated and two downregulated L-ascorbate oxidase homologs were identified in the leaves of *B. papyrifera*, which is compatible with our results about the growth and development requirements of *B. papyrifera* leaves. The cytoskeleton is a highly dynamic protein fiber network structure in cells that regulates the formation of plant cell walls [44]. We identified five downregulated and nine upregulated genes associated with cytoskeleton metabolism from *B. papyrifera* leaves. These results clearly indicated that polysaccharide, cell wall, and cytoskeleton metabolism have an apparent advantage regarding the growth and development of *B. papyrifera* leaves.

To sum up, energy and carbohydrate metabolism, protein and amino acid metabolism, polysaccharide, cell wall, and cytoskeleton metabolism are closely related to the growth and development of *B. papyrifera* leaves and may indirectly affect the synthesis of flavonoids in *B. papyrifera.*

## 4. Materials and Methods

### 4.1. Plant Materials

The leaves (T1 young leaves, T3 immature leaves, and T4 mature leaves) of annual *Broussonetia papyrifera* at different stages were sheared and perforated. They were promptly frozen in liquid nitrogen and kept at –80 °C until they were utilized for RNA extraction and flavonoid content assessment.

### 4.2. Determination of Total Flavonoids

The flavonoids were extracted and quantified using a modified version of Krizek et al.’s (1998) [45] technique. After being ground into homogenate in 2 mL of concentrated HCl: ethanol (1:99, *V*/*V*) mixture, eight 6-mm diameter discs were centrifuged at 12,000× *g* for 10 min. The supernatant was measured at 270, 300, and 330 nm after being soaked in water at 80 °C for 10 min. At each wavelength, the flavonoid concentration was determined using an extinction coefficient of 33 cm^−1^ mM^−1^.

### 4.3. RNA Extraction, cDNA Preparation, and RNA-Seq

According to the manufacturer’s instructions, RNA was extracted using TRIzol (Invitrogen, USA). RNA integrity was determined using the RNA Nano 6000 Assay Kit of the Bioanalyzer 2100 system (Agilent Technologies, CA, USA). The following study was performed on RNA integrity number (RIN) ≥ 7 samples. Total RNA was purified using poly-T oligo-attached magnetic beads to isolate mRNA. At increased temperatures, fragmentation was performed in First Strand Synthesis Reaction Buffer utilizing divalent cations (5X). After first strand cDNA was generated using a random hexamer primer and M-MuLV Reverse Transcriptase, RNA was degraded using RNaseH. Then, the second strand of cDNA was synthesized using DNA Polymerase I and dNTP. PCR was then conducted using Phusion High-Fidelity DNA Polymerase, Universal PCR primers, and Index (X) Primer. The libraries were generated using an Illumina TruSeq Stranded mRNA LT Sample Prep Kit (San Diego, CA, USA) using the instructions provided by the manufacturer. To produce 150 bp paired-end reads, the libraries were sequenced using an Illumina sequencing platform (HiSeq™ 2500). Finally, PCR products were purified using the AMPure XP system, and the quality of the library was assessed using the Qubit2.0 Fluorometer, Agilent Bioanalyzer 2100 system, and qRT-PCR.

### 4.4. Transcript Assembly and Gene Expression Analysis

Raw data were cleaned by eliminating adapter reads, N-base reads, and low-quality reads. The remaining files from all libraries/samples were merged into a single big left.fq file, while the right files were merged into a single large right.fq file. The assembly of the transcriptome was conducted using Trinity [46]. By default, min_kmer_cov was 2, as well as the other parameters. The DESeq2 R software (1.20.0) was used to conduct differential expression analysis on two groups (three biological replicates per group). Their functional categories were assigned according to Nr, Nt, Pfam, KOG/COG, Swiss-Prot, KO, and GO databases.

### 4.5. Quantitative Real-Time PCR (qRT-PCR)

As previously mentioned, total RNA was isolated from the leaves. Three biological duplicates were made for each stage. In Appendix A, the primers used for qRT-PCR are listed. As a control, *Actin* expression was used, with each 20 μL reaction solution comprising 10 μL SYBR Premix Ex Taq mix (Vazyme, Nanjing, China). Then, the following amplification protocol was utilized: initial denaturation at 95 °C for 30 s, then 40 denaturation cycles at 95 °C for 10 s, and 30 s of annealing at 60 °C. The 2^−ΔΔCt^ technique was utilized to determine relative expression degrees, and 3–5 replicates were included in each study.

## 5. Conclusions

Flavonoids are plentiful in the leaves of *Broussonetia papyrifera*, and their manufacture is regulated by numerous genes. Using RNA-seq, we identified 2447 upregulated and 2960 downregulated differentially expressed genes (DEGs), 4657 upregulated and 4804 downregulated DEGs, and 805 upregulated and 484 downregulated DEGs for T1 vs. T3, T1 vs. T4, and T3 vs. T4, respectively. The aforementioned results were integrated with previously published data to explore the mechanism of flavonoid production and control in leaves of *B. papyrifera* at different phases of development. Biosynthesis, modification, transport, and control of flavonoids in plants use both shared and distinct processes. Several important enzyme genes involved in flavonoid production pathways have been discovered, which may explain variations in flavonoid concentration in the leaves of *B. papyrifera*. The results demonstrated that the dynamic changing trend of flavonoids content is related to the expression pattern of the vast majority of essential genes in the biosynthetic pathway. It is believed that these genes are essential for the production of flavonoids in *B. papyrifera* leaves. The flavonoid concentration of *B. papyrifera* leaves may be influenced by genes involved in energy and carbohydrate metabolism, polysaccharide, cell wall and cytoskeleton metabolism, signal transduction, and protein and amino acid metabolism during leaf growth and development (Figure 11). First, we investigated the transcriptome features of flavonoid production in the leaves of *B. papyrifera*. This study’s results offer a foundation for revealing new knowledge about the metabolic pathways of this species. These results may give important insights into the molecular processes that govern flavonoid content and may be applicable to the molecular breeding and development of *B. papyrifera* plant-specific germplasm resources. Moreover, these results give essential information for the improvement of *B. papyrifera* quality.

## Figures and Tables

**Figure 1 plants-12-00563-f001:**
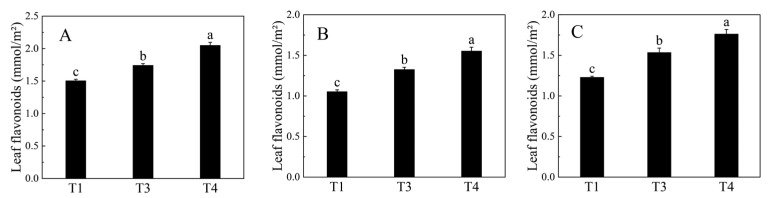
Changes of flavonoids content during leaves development in *Broussonetia papyrifera* at 270 nm (**A**), 300 nm (**B**), and 330 nm (**C**). Three developmental stages were examined in this study as follows: T1: young leaves, T3: immature leaves, T4: matured leaves. Data are means ± SE (*n* = 3). Different letters above the bars indicate a significant difference at *p* < 0.05.

**Figure 2 plants-12-00563-f002:**
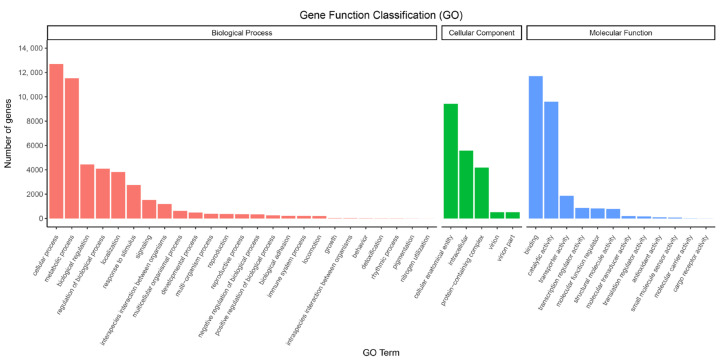
Gene Ontology (GO) function classification of *B. papyrifera* transcriptome. All unigenes were annotated in three functional GO categories: biological process (BP), cellular component (CC) and molecular function (MF).

**Figure 3 plants-12-00563-f003:**
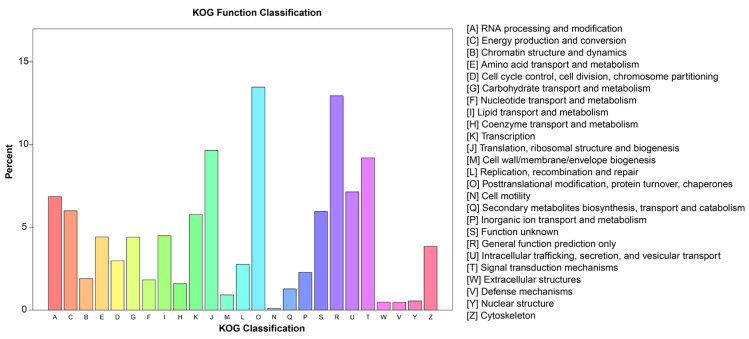
EuKaryotic clusters of Orthologous Groups (KOG) function classification of *B. papyrifera* unigenes. All putative unigenes were analyzed using the KOG database. KOG classifications were divided into 25 functional categories. A total of 8480 unigenes were clustered into 25 KOG classifications.

**Figure 4 plants-12-00563-f004:**
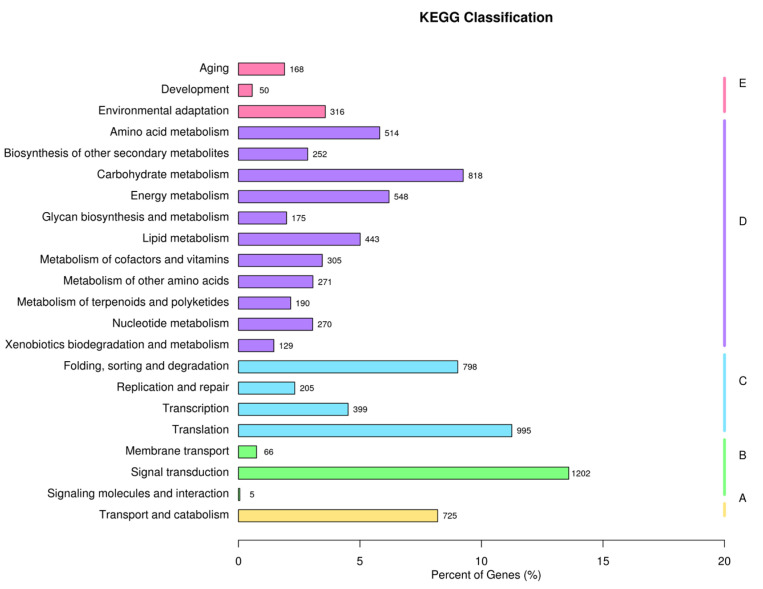
Kyoto Encyclopedia of Genes and Genomes (KEGG) pathway classification of *B. papyrifera* unigenes. The genes were divided into four branches according to the KEGG metabolic pathways: cellular processes, environmental information processing, genetic information processing, metabolism, and organismal systems.

**Figure 5 plants-12-00563-f005:**
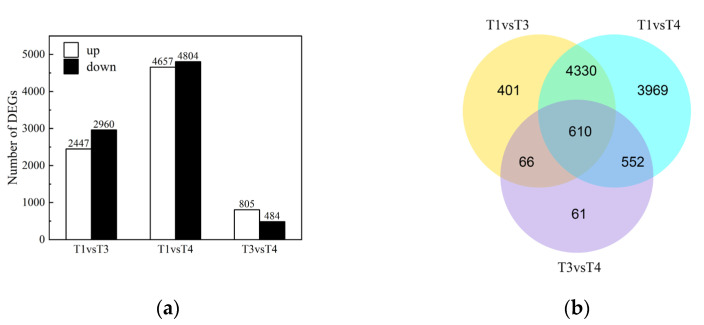
The number of up and downregulated genes in T1 vs. T3, T1 vs. T4, T3 vs. T4 (**a**). Venn diagram showing the number of commonly and uniquely expressed genes in T1 vs. T3, T1 vs. T4, T3 vs. T4 (**b**). (T1: young leaves, T3: immature leaves, T4: matured leaves).

**Figure 6 plants-12-00563-f006:**
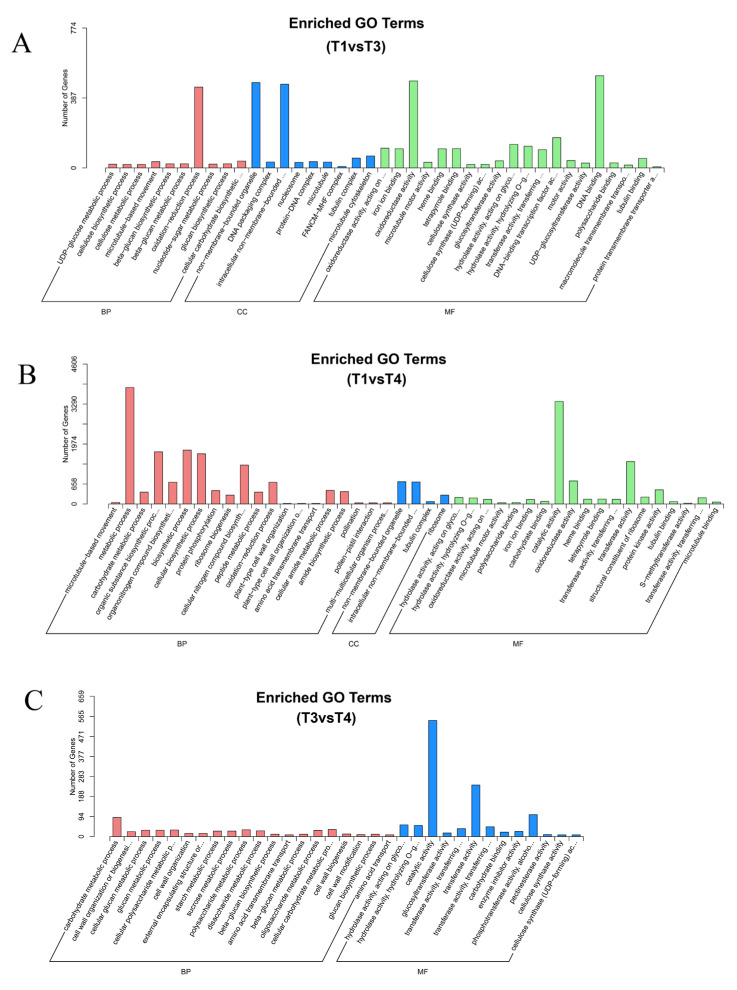
GO classification of differentially expressed genes (DEGs) in T1 vs. T3 (**A**), T1 vs. T4 (**B**), T3 vs. T4 (**C**) (T1: young leaves, T3: immature leaves, T4: matured leaves).

**Figure 7 plants-12-00563-f007:**
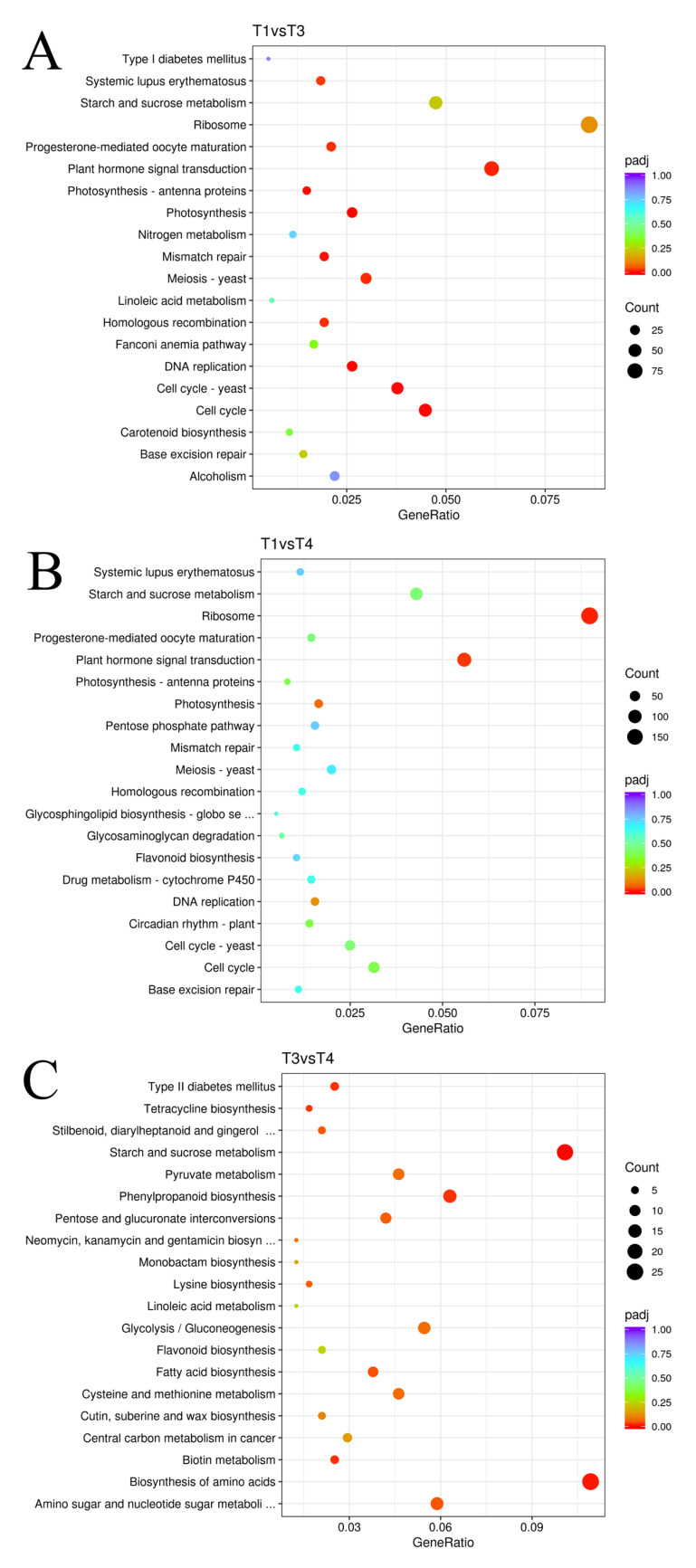
Top 20 enriched KEGG pathway among the annotated DEGs in different leaf stages T1 vs. T3 (**A**), T1 vs. T4 (**B**), T3 vs. T4 (**C**). The degree of enrichment was measured by rich factor, FDR value and the number of genes enriched to each KEGG term. Rich factor refers to the ratio of the number of DEGs enriched in each term to the number of all DEGs annotated. The greater the rich factor, the greater the enrichment. Generally, the value range of FDR is 0–1. The closer to zero, the greater the enrichment. The circle size refers to the number of genes enriched to each KEGG term (T1: young leaves, T3: immature leaves, T4: matured leaves).

**Figure 8 plants-12-00563-f008:**
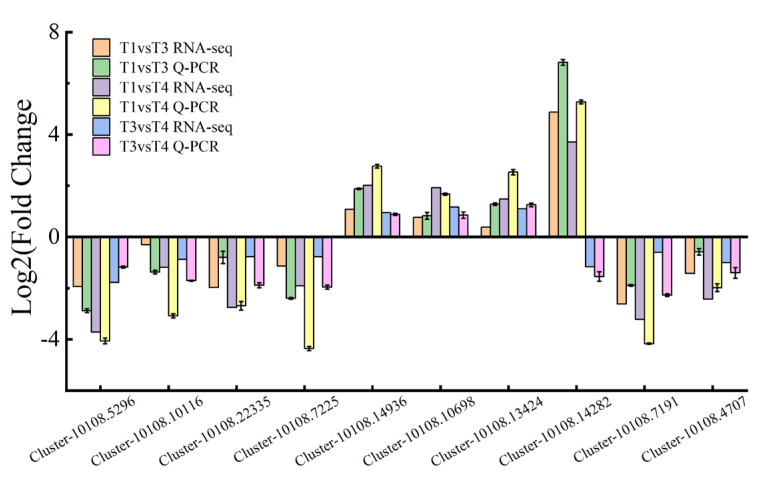
Relative expression levels of DEGs from *Broussonetia papyrifera* leaves at T1, T3, T4 stages. Bars represent means ± SE (*n* = 3). *Actin* was elected as the internal standards and the T1 stage was used as reference sample, which was set to 1. (T1: young leaves, T3: immature leaves, T4: matured leaves).

**Figure 9 plants-12-00563-f009:**
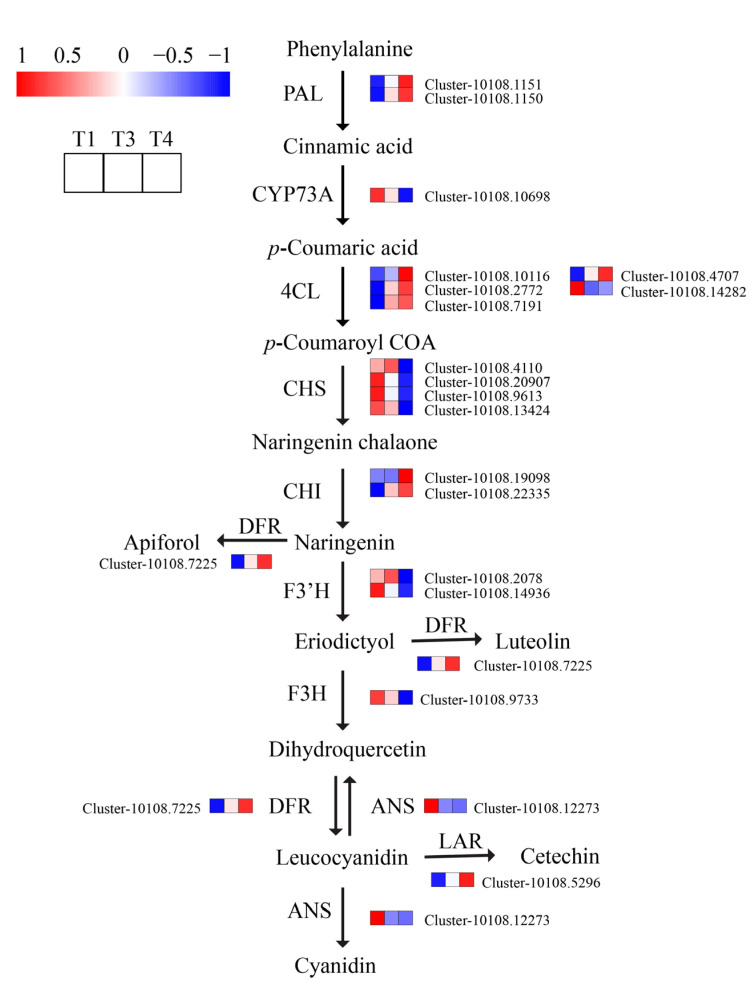
Reconstruction of flavonoid biosynthetic pathway with the differentially expressed genes. PAL: Phenylalanine ammonia-lyase, 4CL: 4-Coumarate-CoA ligase, CHS: Chalcone synthase, CHI: Chalcone-flavonone isomerase, F3H: Flavanone-3-hydroxylase, F3′H: Flavonoid 3′-hydroxylase, DFR: Dihydroflavonol-4-reductase, ANS: Anthocyanin synthase (T1: young leaves, T3: immature leaves, T4: matured leaves).

**Figure 10 plants-12-00563-f010:**
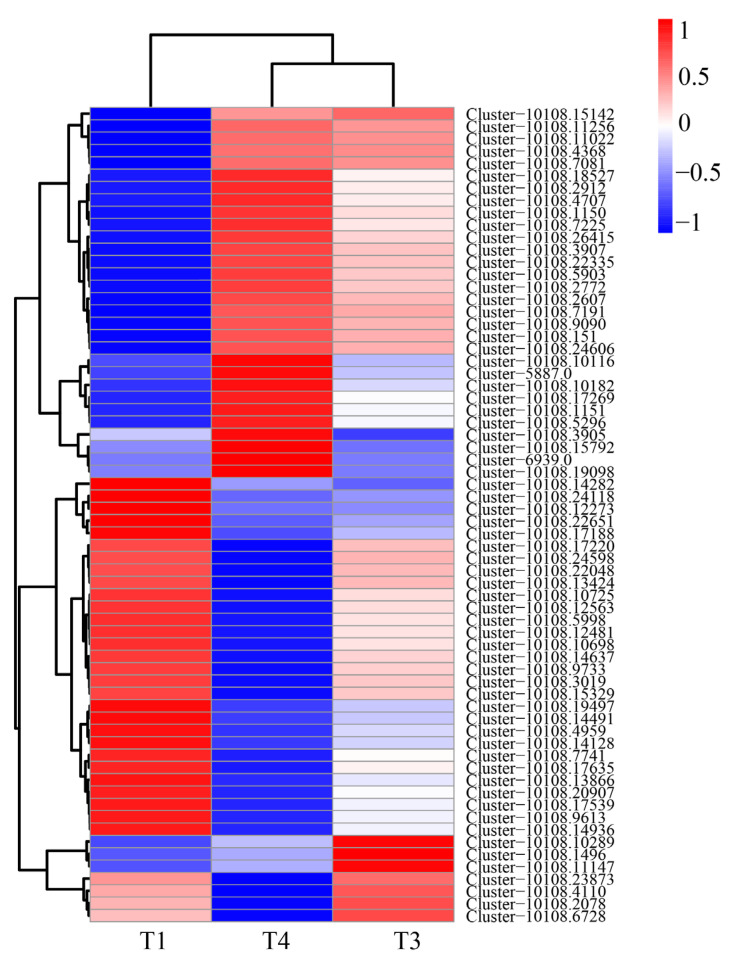
Heat map of differentially expressed genes (DEGs) related to flavonoid biosynthetic pathways at three stages according to hierarchical cluster analysis. Blue indicates the lowest expression; white indicates intermediate expression and red indicates the highest expression. A colour scale bar is shown at the top-right comer of the figure and corresponds to the values of the mean-centred log_2_-transformed fragments per kilobase per million reads (FPKM). (T1: young leaves, T3: immature leaves, T4: matured leaves).

**Figure 11 plants-12-00563-f011:**
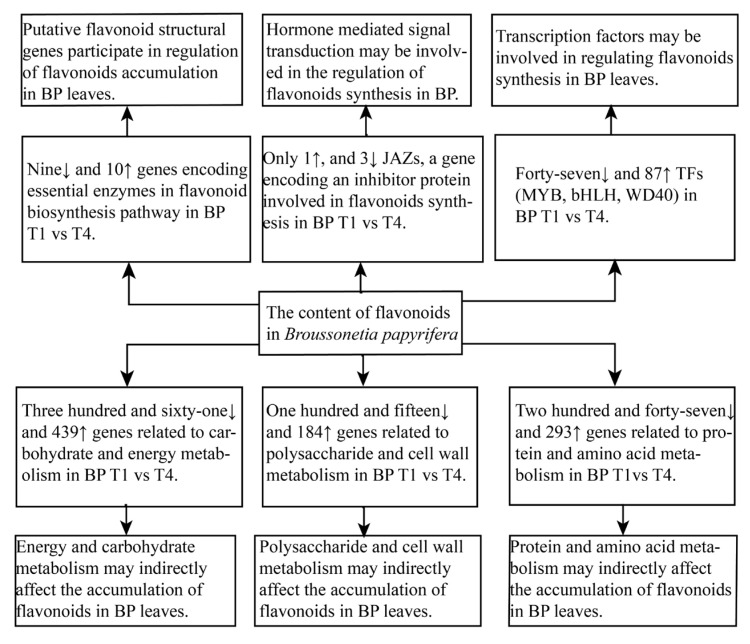
Potential regulatory pathways affecting flavonoids accumulation in *Broussonetia papyrifera* leaves. BP, *Broussonetia papyrifera*; JAZ, Jasmonate Zim domain; MYB, v-myb avian myeloblastosis viral oncogene homolog; bHLH, basic helix-loop-helix; WD40, WD-repeat protein (T1: young leaves, T3: immature leaves, T4: matured leaves).

**Table 1 plants-12-00563-t001:** Summary of sequencing quality.

Sample ^1^	Raw Reads	Clean Reads	Clean Bases	Error (%)	Q20 (%)	Q30 (%)	GC Content (%)
T1-1	21,325,528	20,725,009	6.22G	0.03	97.84	93.76	46.63
T1-2	22,271,891	21,756,316	6.53G	0.03	97.75	93.61	46.76
T1-3	21,793,119	21,357,102	6.41G	0.03	97.82	93.76	46.23
T3-1	22,174,936	21,643,438	6.49G	0.03	97.60	93.30	46.16
T3-2	22,126,417	21,170,390	6.35G	0.03	97.87	94.00	46.63
T3-3	21,308,885	20,786,945	6.24G	0.03	97.97	94.11	46.71
T4-1	23,421,640	22,539,743	6.76G	0.03	97.89	93.95	46.15
T4-2	22,434,529	21,668,818	6.50G	0.03	97.88	94.03	46.10
T4-3	22,656,272	22,197,669	6.66G	0.03	97.48	92.79	45.86

^1^ Leaves of *Broussonetia papyrifera* at different stages (T1: young leaves, T3: immature leaves, T4: matured leaves).

**Table 2 plants-12-00563-t002:** The length distribution of Unigenes and transcripts about *Broussonetia papyrifera*.

	300–500 bp	500 bp–1 kbp	1 kbp–2 kbp	>2 kbp	Total
Number of Unigenes	11,342	10,629	7915	11,651	41,537
Number of transcripts	17,930	20,939	32,898	52,898	124,575

**Table 3 plants-12-00563-t003:** Summary of the assembly result.

	Min Length	Mean Length	Median Length	Max Length	N50	N90	Total Nucleotides
Transcripts	301	2005	1705	17,100	2835	1078	249,817,198
Genes	301	1515	902	17,100	2568	592	62,948,261

**Table 4 plants-12-00563-t004:** Summary of unigenes function annotation results.

Datebase	Number of Unigenes	Percentage
NR	25,804	62.12
NT	19,586	47.15
KO	10,361	24.94
Swiss Prot	22,354	53.81
PFAM	21,289	51.25
GO	21,287	51.24
KOG	8480	20.41
Annotated in all Databases	3358	8.08
Annotated in at least one Database	30,177	72.65
Total Unigenes	41,537	100

**Table 5 plants-12-00563-t005:** Fold change result of DEGs in leaves of *Broussonetia papyrifera*.

Compare *	|log2FoldChange| < 2	Percentage	|log2FoldChange| > 2	Percentage
T1vsT3	3190	59.00%	2217	41.00%
T3vsT4	874	67.80%	415	32.20%
T1vsT4	4272	45.15%	5189	54.85%

* Leaves of *Broussonetia papyrifera* at different stages (T1: young leaves, T3: immature leaves, T4: matured leaves).

## Data Availability

All data are available in the manuscript or the Appendix A. The raw RNA-Seq reads are available at the NCBI Sequence Read Archive (SRA): PRJNA806384.

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
