# Peer review of "Transcriptome Sequencing of Broussonetia papyrifera Leaves Reveals Key Genes Involved in Flavonoids Biosynthesis"

_plants, 2023, doi:10.3390/plants12030563_

Round 1

Reviewer 1 Report

The manuscript is focused on the rich in flavonoids Broussonetia papyrifera. Three leaf developmental stages are compared in respect to their flavonoid content and transcriptome. The research could highlight the differential expression of genes involvement in the flavonoid biosynthesis and regulation of flavonoid level. Indeed, the study is a strong platform for future research into B. papyrifera's metabolic pathways. In my opinion, the work will be of interest for a broad range of scientists.

The manuscript is written concisely, and the work is represented clearly in all the components that are included in: Introduction, Results with Figures and Tables, Discussion, Material and methods.

Comments:

-Page 1, line 30 – the Latin name in the beginning of the sentence should be fully written.

-The link (PlnTFDB: http://plntfdb.bio.uni-potsdam.de/ v3.0/) is not available. Could you update, or cite the article?

-Figure 8 is now labelled as Figure 7 in the legend.

-Figure 9 is now labelled as Figure 8 in the legend.

-Check figure numbering in Discussion as well.

-What standard was used for measurement of flavonoid content? Please, add this information to Material and Methods.

-Where are deposited the RNA-Seq data? Please, make this information available.

Author Response

请参阅附件。

Reviewer 2 Report

Review of “Transcriptome sequencing of Broussonetia papyrifera leaves reveals key genes involved in flavonoids biosynthesis

Basic reporting

This article agrees with Plants scope.

The transcriptional control of polyphenol in Broussonetia papyrifera is an important issue for Broussonetia papyrifera breeding selection programs as well as for recovering general information extensive to other crops, for which I consider the objective of this research to be widely justified.

The research has been conducted rigorously and methods are detailed described. The structure of the article is well organized according to the journal standard sections.

In introduction section, a large explanation of flavonoids biosynthesis is addressed, maybe further than necessary. Background and cited literature are sufficient and appropriate to frame the research.

In results, a very short explanation of “Changes in total flavonoid content during B. papyrifera leaves development” is provided. The reviewer recommends the authors to explain the three absorbances differences founded and a deep explanation of types of flavonoids founded, as all the results are aimed to explain flavonoid changes. Just as suggestion, a more accurate description of flavonoid appearing in leaves in each stage could be revealing.

There is another issue of concern, perhaps due to reviewer ignorance, but…why the KEGG annotation of putative proteins in Broussonetia papyrifera refers to a digestive system, circulatory system, … presumably not concerning to plant systems?

The reviewer is not sure about the coherency of results with initial hypothesis. Author provides a large abundance of results, very complex but well presented. In the other hand, interpreting results independently linking with flavonoid contents seems to be forced. One example is the follow statement “These results clearly indicated that polysaccharide, cell wall, and cytoskeleton metabolism have an apparent advantage on the growth and development of B. papyrifera leaves. Consequently, they may be essential for flavonoid synthesis.” (line 386)

This article has enough information for publication, even for splitting in two; one concerning to flavonoid pathway itself and a general one concerning to development of B. papyrifera. But in the reviewer opinion, there are some processes that take place at same time but not necessarily connected, even if expression data correlates.

Author Response

请参阅附件。

Reviewer 3 Report

It is recommended to rephrase abstract and introduction to be more concise and focus only on information relevant to paper.

Figure 2 and 6 legends are difficult to read.

Specific gene expression data for DEGs form qRT-PCR  between different stages is missing.

Results lack drawing of any critical conclusion and most of the discussion reads like results rather than discussion about the observations.

Author Response

请参阅附件。

Round 2

Reviewer 3 Report

Thank you for the revisions.